# Social mixing patterns in the UK following the relaxation of COVID-19 pandemic restrictions, July–August 2020: a cross-sectional online survey

Jessica RE Bridgen ![ORCID] , Chris Jewell, Jonathan M Read ![ORCID]

Lancaster Medical School, Lancaster University Faculty of Health and Medicine, Lancaster, UK

**Correspondence to**
Dr Jonathan M Read;
jonathan.read@lancaster.ac.uk

## ABSTRACT

**Objectives** To quantify and characterise non-household contact and to identify the effect of shielding and isolating on contact patterns.

**Design** Cross-sectional study.

**Setting and participants** Anyone living in the UK was eligible to take part in the study. We recorded 5143 responses to the online questionnaire between 28 July 2020 and 14 August 2020.

**Outcome measures** Our primary outcome was the daily non-household contact rate of participants. Secondary outcomes were propensity to leave home over a 7 day period, whether contacts had occurred indoors or outdoors locations visited, the furthest distance travelled from home, ability to socially distance and membership of support bubble.

**Results** The mean rate of non-household contacts per person was 2.9 d⁻¹. Participants attending a workplace (adjusted incidence rate ratio (aIRR) 3.33, 95% CI 3.02 to 3.66), self-employed (aIRR 1.63, 95% CI 1.43 to 1.87) or working in healthcare (aIRR 5.10, 95% CI 4.29 to 6.10) reported significantly higher non-household contact rates than those working from home. Participants self-isolating as a precaution or following Test and Trace instructions had a lower non-household contact rate than those not self-isolating (aIRR 0.58, 95% CI 0.43 to 0.79). We found limited evidence that those shielding had reduced non-household contacts compared with non-shielders.

**Conclusion** The daily rate of non-household interactions remained lower than prepandemic levels measured by other studies, suggesting continued adherence to social distancing guidelines. Individuals attending a workplace in-person or employed as healthcare professionals were less likely to maintain social distance and had a higher non-household contact rate, possibly increasing their infection risk. Shielding and self-isolating individuals required greater support to enable them to follow the government guidelines and reduce non-household contact and therefore their risk of infection.

## INTRODUCTION

On 31 January 2020, the first two cases of COVID-19 were recorded in the UK, followed by a rapid rise in identified cases and hospitalised patients. On 23 March 2020, a range of social distancing measures were implemented

**STRENGTHS AND LIMITATIONS OF THIS STUDY**

⇒ Large-scale cross-sectional study.
⇒ This study provides detailed information on non-household contact and associated behaviours.
⇒ The study period corresponds with the start of epidemic growth, behaviours measured could provide insight into the level of social mixing needed to support epidemic growth.
⇒ As there was no active recruitment process, certain demographic groups are under-represented and the study may suffer from recruitment bias.
⇒ Social contacts were self-reported by participants and were therefore subject to recall bias.

across the UK (lockdown), aiming to reduce interpersonal contact between households and reduce transmission of SARS-CoV-2. Schools were closed to pupils, with the exception of children of key workers. People were only allowed to leave their homes to shop for basic necessities, to exercise once a day, for medical reasons and to travel to work if working from home was not possible.[1] By July 2020, many businesses, including shops, restaurants and pubs, had reopened. Support bubbles had been introduced, allowing for a single-adult household to interact with another household of any size.[2] International travel was permitted, following the introduction of travel corridors on 10 July 2020, which enabled passengers to travel to England from certain countries without self-isolating.[3] The UK government's 'Eat Out to Help Out' scheme, which ran from 3 August 2020 to 31 August 2020, encouraged people to dine out.[4] Some social distancing restrictions remained in place, including maintaining a 2 m distance between individuals (excluding household members or members of a support bubble), the wearing of face coverings on public transport and in shops, and limits on how many people could meet indoors and outdoors.[5–8] While some people

in the UK began to return to work, schools remained closed. A marked decrease in case incidence was seen during April 2020, and cases remained low until the onset of the second wave in August 2020.

Epidemics are largely driven by social mixing patterns and their quantification is useful for transmission modelling purposes, as well as assessing adherence to regulations and identifying sociodemographic factors associated with heterogeneities in contact rate.[9–11] The apparent association between social distancing restrictions and reduced case incidence indicates that a nuanced understanding of how individuals' contact patterns vary could inform behavioural interventions for the remainder of the outbreak. Previous contact studies have provided estimates for age-specific contact rates in Great Britain and the UK.[11–13] A cross-sectional survey of UK adults early on during the lockdown beginning in March found a substantial reduction in daily contact between people.[14]

We conducted a cross-sectional online survey between 28 July 2020 and 14 August 2020 to measure the mobility of people living in the UK, which locations people were frequenting and the number of non-household contacts people were making. We aimed to quantify non-household contact behaviour and adherence to self-isolation and shielding guidance. The study period coincided with the start of the second wave of SARS-CoV-2 infection in the UK, when hospital admissions for COVID-19 were at their lowest rate since April.[15]

## METHODS
### Survey methodology
Data collection was conducted through an anonymous online questionnaire; the study was branded the CoCoNet (COVID-19 Contact Network) survey. The survey was open to anyone living in the UK at the time of the survey. There was no lower age limit for participation, with children under 13 required to complete the survey with a parent or guardian. The inclusion criteria for participants were that they completed the question on residency location and that they were resident in the UK at the time of the survey.

The survey was promoted through a university press release, engagement with the media, and posts on social media directing potential participants to the study website: https://www.lancaster.ac.uk/health-and-medicine/research/coconet-study/.

Demographic information from participants, including age, sex, ethnicity, home location (first part of postcode) and their employment or school situation, was collected. Participants were asked about their household size, as well as the formation and size of support bubbles they may belong to. Participants were asked about their activities on the previous day (the contact reporting day), including whether they left their household and the number and characteristics of non-household contacts encountered. The questionnaire is presented in online supplemental material and the dataset is publicly available.[16]

To reduce participant burden, a triage question on how many people participants had met the previous day determined the level of information collected on contacts. Participants reporting fewer than 15 contacts were asked to estimate the age of each contact they made, whether they met the contact indoors or outdoors, and if anyone from their household had also met that contact the same day. Participants who reported 15 or more contacts were asked to estimate the number of contacts made with different age groups, and whether they had met most of their contacts indoors or outdoors.

Responses recorded between 00:00BST 28 July and 18:00BST 14 August 2020 were included in the analysis. Partial responses to the survey were analysed if the first compulsory question asking which part of the UK a participant resided in was answered. If a participant exited the online survey early, we used their responses up to and including the last question they saw.

### Primary and secondary outcome measurements
Our primary outcome was non-household contact rate. A non-household contact was defined as someone with whom the participant had a face-to-face conversation with, excluding members of their own household. A participant who remained at home could still make non-household contacts by having visitors to their home.

Secondary outcomes were whether contacts occurred indoors or outdoors, propensity to leave home over a 7-day period, ability to socially distance, locations visited, furthest distance travelled from home and membership of support bubble.

### Descriptive analysis
Representativeness was assessed by visual comparison of participant demographics with respective Office for National Statistics 2019 mid-year estimates.[17 18] The mean number of non-household contacts was calculated and stratified by age, sex and household size, and was compared with reported values from other social contact surveys. Adherence to social distancing guidance was assessed by calculating the proportion of participants who left home in the past 7 days, the distribution of furthest distance travelled in the past 7 days, and the proportion of participants who felt able to maintain a recommended physical distance during contact with others. Non-responses were excluded from analyses.

### Predictors of contact frequency
To identify characteristics of the participant associated with their rate of daily non-household contact, we fitted a negative binomial model to the daily number of non-household contacts reported by participants. Explanatory variables included in the model a priori were: age; sex; ethnicity; nation of residence (England, Northern Ireland, Scotland or Wales); household size; dwelling type; whether the contact reporting day was a weekend or week day; whether the participant had left their home on the contact reporting day; participant's working situation;

participant's COVID-19 circumstance. To support our hypothesis-driven choice of model parameters, we also conducted a forward stepwise model selection process, with our previously selected explanatory variables used as candidate variables (see online supplemental materials). Statistical analyses were conducted by using R V.4.0.2.[19]

## Patient and public involvement statement

Patients or the public were not involved in the design, or conduct, or reporting, or dissemination plans of our research. However, as the online survey was promoted via social media, members of the public were free to further promote it via social media links.

## RESULTS

### Participant demographics

We received 5383 survey responses recorded between 28 July 2020 and 14 August 2020; 5143 responses met our inclusion criteria.[16] Most participants were aged 40–59 (55.3%, 2813/5090) (table 1, figure 1A). We recorded fewer responses from participants in the youngest age groups, 0–9 years (0.1%, 5/5090) and 10–19 years (0.7%, 38/5090) and in the oldest age group, aged 80+ (0.4%, 21/5090). Males, non-white ethnicities and residents of Northern Ireland and Wales were under-represented in our sample.

### Mobility

We found 33.7% (95% CI 32.4% to 35.0%) of participants left their home every day over a 7-day period (table 2). Over the same time period, most participants travelled less than 10 miles from home, but some longer-range travel (50+ miles) occurred.

### Non-household contacts

A total of 14388 non-household contacts were recorded by 5037 participants. The mean rate of non-household contacts was 2.9 $d^{-1}$ (95% CI 2.7 to 3.0). This is a notably lower rate of non-household contact than recorded from prepandemic surveys (online supplemental table 1). We found 33.4% (95% CI 32.1% to 34.7%) of participants made no non-household contacts. The degree distribution of non-household contacts has a long right-hand tail (95th percentile: 10 contacts $d^{-1}$, maximum 130 contacts $d^{-1}$); figure 1B. We also quantified the non-household contact rate of household members of participants (online supplemental materials).

Mean non-household contact rate varied by age and was highest among 10–19 years (mean 3.6, 95% CI 1.6 to 6.5) (figure 1C). We found moderate assortative mixing by age, in line with both current and prepandemic contact studies (online supplemental figure 1A). We found that the mean daily non-household contact rate by participant age group was substantially lower when compared with prepandemic POLYMOD study (see online supplemental figure 1B). A notable decrease in contact rate was found between people aged under 60 mixing with others aged under 60, with the largest reduction in contact rate seen across all age groups when mixing with 0–19 years (online supplemental figure 1B).

**Table 1** Participant demography and UK ONS 2019 mid-year estimates

| | No. of participants (%) | UK ONS mid-year estimates (2019)* (%) |
|---|---|---|
| Age group (N=5090)† | | |
| 0–9 | 5 (0.1) | 12.0 |
| 10–19 | 38 (0.7) | 11.4 |
| 20–29 | 256 (5.0) | 13.0 |
| 30–39 | 598 (11.7) | 13.3 |
| 40–49 | 1183 (23.2) | 12.6 |
| 50–59 | 1630 (32.0) | 13.6 |
| 60–69 | 1065 (20.9) | 10.7 |
| 70–79 | 294 (5.8) | 8.4 |
| 80+ | 21 (0.4) | 5.0 |
| Sex (N=5090)† | | |
| Female | 4017 (78.9) | 50.6 |
| Male | 1051 (20.6) | 49.4 |
| Prefer not to say | 22 (0.4) | – |
| Ethnicity (N=5090) | | |
| White | 4880 (95.9) | 86.0 |
| Mixed/Multiple ethnic groups | 49 (1.0) | 2.2 |
| Asian/Asian British | 50 (1.0) | 7.5 |
| Black/African/Caribbean/ Black British | 11 (0.2) | 3.3 |
| Other ethnic groups | 7 (0.1) | 1.0 |
| Prefer not to say | 16 (0.3) | – |
| No response | 77 (1.5) | – |
| Nation (N=5143)† | | |
| England | 4714 (91.7) | 84.3 |
| Northern Ireland | 33 (0.6) | 2.8 |
| Scotland | 254 (4.9) | 4.7 |
| Wales | 142 (2.8) | 8.2 |
| Household size (N=5073)† | | |
| 1 | 878 (17.3) | 29.5 |
| 2 | 1911 (37.7) | 34.5 |
| 3 | 987 (19.5) | 15.4 |
| 4 | 907 (17.9) | 13.9 |
| 5 | 287 (5.7) | 4.5 |
| 6+ | 103 (2.0) | 2.1 |

N is the number of participants who provided a response to the question.
*Ethnicity estimates from 2011 census data.
†Question required a response from participants to progress through the online survey.
ONS, Office for National Statistics.

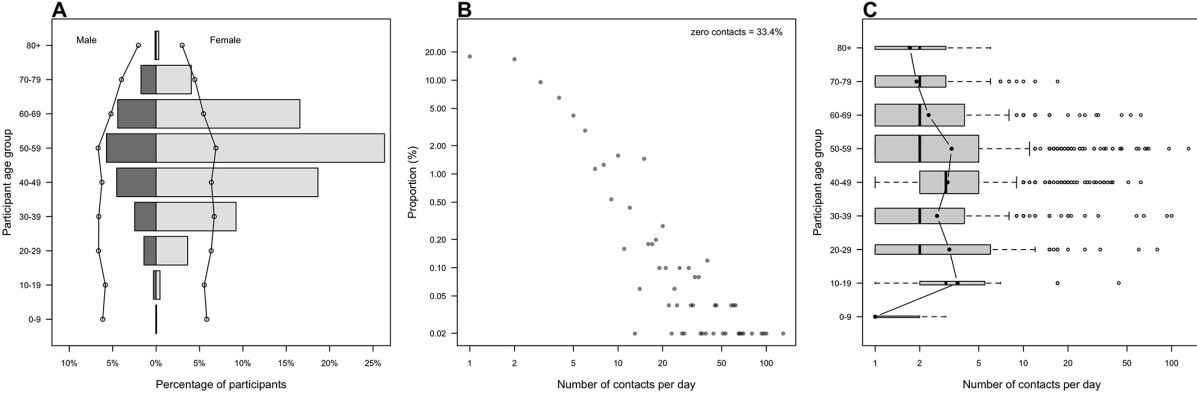

**Figure 1** (A) Age and sex distribution of participants with ONS 2019 mid-year estimates. (B) Degree distribution of non-zero contacts. (C) Distribution of reported non-zero contact rate by age group. Note, log scale of x-axis in (B, C). ONS, Office for National Statistics.

## Participant characteristics and non-household contact rate

We identified the association of participant characteristics with the rate of non-household contact using a multiple regression model (figure 2, online supplemental table 2). The candidate variable dwelling type was not selected by the model selection process (online supplemental table 3). We found no association of non-household contact rate with sex or day of the week. Contact rate varied by participant age: participants aged 30–39 (adjusted incidence rate ratio (aIRR) 0.86, 95% CI 0.76 to 0.97), aged 40–49 (aIRR 0.90, 95% CI 0.82 to 1.00) and those aged 60–69 (aIRR 0.89, 95% CI 0.79 to 1.00) reported a lower rate of contact than participants aged 50–59. We found that Asian and Asian British participants had a lower rate of contact than White participants (aIRR 0.54, 95% CI 0.36 to 0.82). Participants residing in Scotland had a lower contact rate than those living in England (aIRR 0.80, 95% CI 0.68 to 0.95), whereas participants in Wales had a higher contact rate (aIRR 1.22, 95% CI 0.99 to 1.50).

Leaving home was associated with a higher non-household contact rate than staying at home (aIRR 5.58, 95% CI to 4.92 to 6.33). Attending a workplace (aIRR 3.33, 95% CI 3.02 to 3.66), being self-employed (aIRR 1.63, 95% CI 1.43 to 1.87) or working in healthcare (aIRR 5.10, 95% CI 4.29 to 6.10) was associated with a significantly higher rate of non-household contact than working at home.

## Social distancing characteristics of shielding and self-isolating individuals

There were 353 (6.9%, 353/5073) participants who reported their COVID-19 circumstance to be shielding, either due to being a vulnerable individual or living with a vulnerable individual. In addition, 136 (2.7%, 136/5073) participants reported their COVID-19 circumstance as self-isolating. Shielding individuals tended to be older than non-shielding individuals (online supplemental table 4).

Shielding and self-isolating participants were less likely to leave their home compared with those reporting their situation to be 'not self-isolating or shielding': 58.6% (95% CI 53.2% to 63.8%) of shielding individuals, 52.6% (95% CI 43.8% to 61.2%) of self-isolating individuals and 82.7% (95% CI 81.6% to 83.8%) of other participants reported leaving their home during the contact day (online supplemental table 4). The majority of shielding and self-isolating participants adhered to contemporary social distancing guidelines: 70.1% (95% CI 62.5% to 76.9%) of shielding participants and 73.6% (95% CI 59.7% to 84.7%) of self-isolating participants reported maintaining social distance at all time with contacts met the previous day (online supplemental table 4).

Shielding and self-isolating individuals made fewer contacts per day outside of the household than non-shielding or isolating individuals. The unadjusted rate of non-household contact was 1.3 d⁻¹ (95% CI 1.1 to 1.5) among shielding participants, 1.2 d⁻¹ (95% CI 0.7 to 2.1) for self-isolating participants and 3.1 d⁻¹ (95% CI 2.9 to 3.2) for participants who were not self-isolating or shielding. After adjusting for other variables, we found vulnerable individuals shielding had a marginally lower non-household contact rate than those not shielding or self-isolating (aIRR 0.82, 95% CI 0.66 to 1.01). Those self-isolating as a precaution or under test and trace instructions had a lower non-household contact rate than individuals not shielding or self-isolating (aIRR 0.58, 95% CI 0.43 to 0.79) (figure 2). Individuals who reported as self-isolating with symptoms had a higher rate of non-household contact than those not self-isolating or shielding (aIRR 4.05, 95% CI 1.94 to 9.72). However, a single participant in this group reported a very large number of contacts on their contact day. This is not necessarily an example of non-adherence to social distancing guidance, as contact day and current day are different days. Our questionnaire design asked about contact on the day prior to completing the survey, which would be the day of their current COVID-19 situation. When we exclude this individual from our analysis, we found no significant difference in contact rate (see online supplemental table 5).

**Table 2** Ability of participants to social distance, membership and size of support bubbles, locations visited and mobility of participants

| | No of participants (%) |
|---|---|
| Maintaining social distance yesterday (N=3249)* | |
| All of the time | 1910 (58.8) |
| More than half of the time | 934 (28.7) |
| Less than half of the time | 296 (9.1) |
| None of the time | 89 (2.7) |
| Not sure | 20 (0.6) |
| Part of a support bubble (N=5066)* | |
| Yes | 2029 (40.1) |
| No | 3037 (59.9) |
| Support bubble size (N=2011) | |
| 1 | 866 (43.1) |
| 2 | 560 (27.8) |
| 3 | 229 (11.4) |
| 4 | 201 (10.0) |
| 5+ | 155 (7.7) |
| No response | 18 |
| Frequency of leaving home in past 7 days (N=4896) | |
| 0 days | 82 (1.7) |
| 1 day | 281 (5.7) |
| 2 days | 518 (10.6) |
| 3 days | 605 (12.4) |
| 4 days | 568 (11.6) |
| 5 days | 650 (13.3) |
| 6 days | 537 (11.0) |
| 7 days | 1650 (33.7) |
| Not sure | 5 (0.1) |
| No response | 30 |
| Locations visited yesterday (N=4034) | |
| Someone's home | 615 (15.2) |
| School or workplace | 612 (15.2) |
| Doctor's surgery or healthcare facility | 182 (4.5) |
| Supermarket or convenience store | 1473 (36.5) |
| Other shops or retail spaces | 596 (14.8) |
| Restaurant, café or pub | 553 (13.7) |
| For a walk or exercise | 2178 (54.0) |
| Other | 808 (20.0) |
| No response | 0 |
| Furthest distance travelled in past 7 days (N=4913) | |
| Under two miles | 886 (18.0) |
| 2–9 miles | 1682 (34.2) |
| 10–19 miles | 848 (17.3) |
| 20–49 miles | 669 (13.6) |

Continued

**Table 2** Continued

| | No of participants (%) |
|---|---|
| 50+ miles | 828 (16.9) |
| No response | 13 |

N is the number of participants who provided a response to the question.
*Question required a response from participants to progress through the online survey.

## Ability to maintain social distancing

Participants were asked how much of the time they were able to maintain social distance from everyone they had met the previous day, excluding members of their household and support bubble. We found 58.8% (95% CI 57.1% to 60.5%) of participants felt able to maintain social distancing at all times, while 2.7% (95% CI 2.2% to 3.4%) felt unable to maintain social distance at any time. We found that age and employment situation were associated with being able to 'maintain social distance more than half of the time' (online supplemental table 6.) Participants aged 30–39 felt less able to maintain social distance more than half of the time compared with 50–59 years (adjusted OR (aOR) 0.66, 95% CI 0.46 to 0.95). Healthcare professionals (aOR 0.26, 95% CI 0.17 to 0.40) and those attending their workplace in-person (aOR 0.71, 95% CI 0.53 to 0.96) were less likely to be able to maintain social distance than those working from home.

## Location of encounters

Transmission risk of SARS-CoV-2 is thought to be greater in enclosed, non-ventilated spaces and lower in outdoor environments.[20] To assess how interactions may be distributed by these settings, we asked participants reporting fewer than 15 individual contacts whether each contact was made indoors or outdoors, and asked all participants if they met all or the majority of contacts indoors or outdoors. The distribution of contacts by indoor/outdoor setting was bimodal: nearly half of participants reported meeting all of their non-household contacts indoors (48.8%, 95% CI 47.0% to 50.6%), while 33.7% (95% CI 32.1% to 35.4%) of participants reported meeting all of their non-household contacts outdoors. We also explored the non-household contacts of participants that remained at home (visitors) and the characteristics associated with visiting another household (see online supplemental table 7).

## DISCUSSION

We found the daily rate of social contact was considerably lower than that measured prior to 2020 in similar but non-identical studies, despite our study period corresponding to a time when the COVID-19 pandemic social distancing restrictions were at their most relaxed during 2020 in

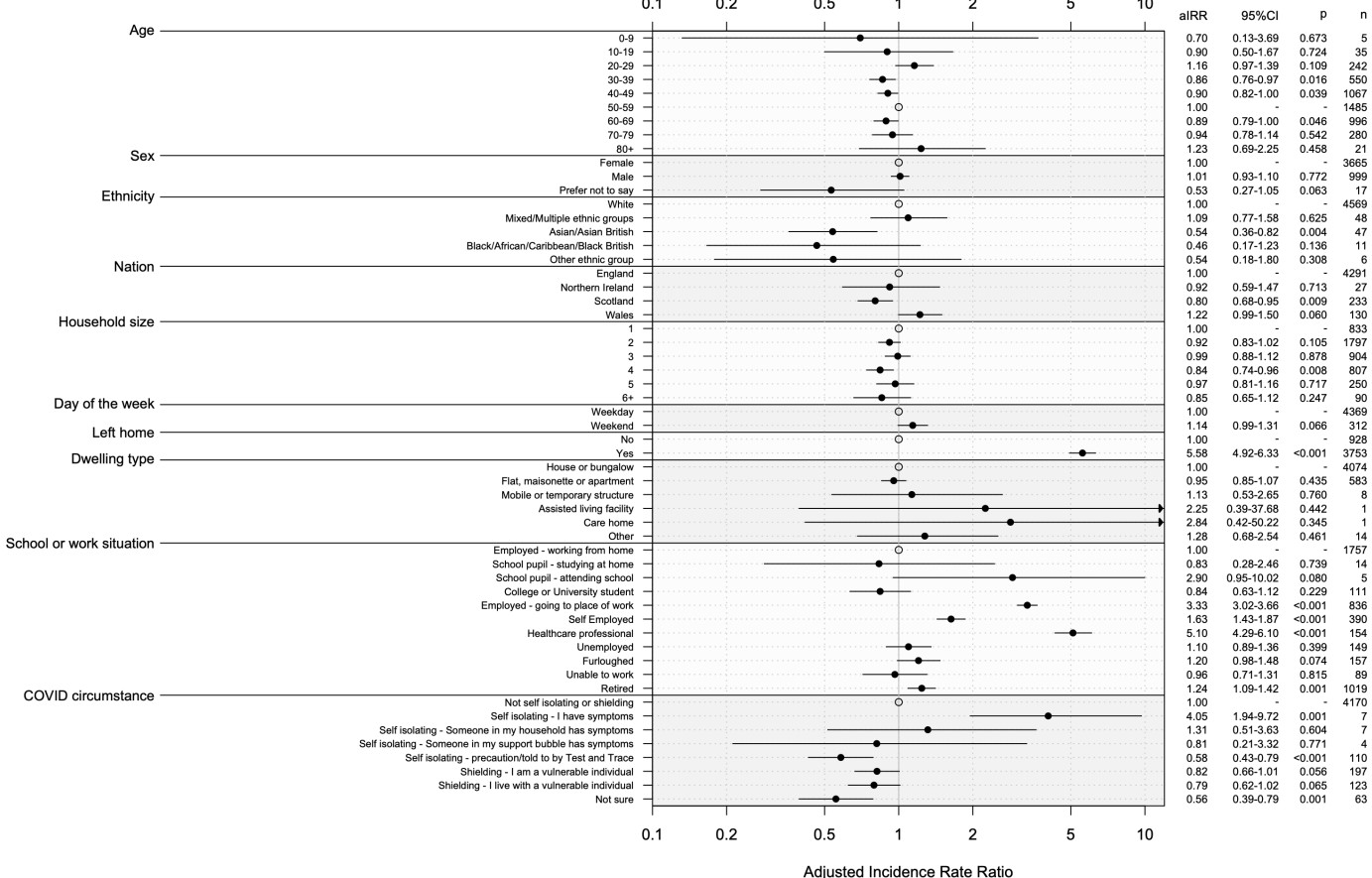

**Figure 2** Adjusted incidence rate ratios for number of non-household contacts reported for selected variables.

the UK.[11–13 21] The Comix study of UK social contact rates reported a greatly reduced rate in March 2020 which increased during summer 2020, with the highest rate of contact recorded in August remaining markedly lower than prepandemic contact rate estimates.[22] Social contact studies outside of the UK also reported low daily contact rates in 2020.[23–25] A similar increase in contact rate following lockdown was observed by Latsuzbaia et al in Luxembourg.[26]

Contact rates and ability to follow social distancing guidelines were associated with age and occupation. The older age groups (70–79, 80+), those at highest risk of severe COVID-19 outcomes, had the lowest non-household contact rates, and they mixed most often with 20–59 years. Individuals attending a workplace, or those self-employed or working in healthcare, had a higher daily non-household contact rate than those working from home, representing additional potential infection risk. A small proportion of participants reported making a large number (more than 50) of non-household contacts; these were exclusively participants who reported their employment situation as either attending their workplace in-person or working as a healthcare professional. Although the UK government was encouraging people to return to work at this time, we found that a high proportion of employed individuals (70.0%, excluding healthcare workers and those self-employed) continued to

work from home.[27] In contrast to prepandemic contacts surveys, we found no association between non-household contact rate and day of the week.[11 13]

Black and Asian individuals have been shown to be at increased risk of SARS-CoV-2 infection in comparison to White individuals, possibly due to larger households, being more likely to be employed as essential workers, and less able to work from home.[28 29] However, after accounting for home-working, we found that individuals of Asian and Asian British ethnicity had a significantly lower non-household contact rate than White participants. This suggests that workplaces may be more dominant as a source of infection for these individuals than previously thought.[30]

The majority of participants reported being able to maintain social distance from others more than half of the time and very few participants reported failing to maintain social distance at all, a similar observation made in a UK behavioural cohort.[31] Healthcare professionals and employees attending their workplace in-person were less able to maintain physical distance from people they encountered than people working from home. This highlights the increased risk of infection that some workers may face; occupations which require employees to interact closely with a large number of people are associated with an increased likelihood of exposure to COVID-19 and clusters of cases developing at a workplace.[32–34]

We found some evidence of non-adherence to self-isolating and shielding guidelines, with a high proportion of self-isolating and shielding participants leaving their home the previous day. Smith *et al* also found low adherence to isolation instructions among the UK population during March–August 2020.[35] We found that a large proportion of self-isolating and shielding participants (including those living with vulnerable individuals) made non-household contacts, suggesting shielding and self-isolating individuals needed greater support to further reduce their number of interactions and to minimise infection risk.

Participants who were self-isolating as a precautionary measure, or after having been contacted by Test and Trace, reported fewer contacts than those not shielding or self-isolating. However, participants self-isolating due to experiencing symptoms or when a member of their household had symptoms did not have reduced contact rate, possibly due to the small number of participants reporting these circumstances. Participants who reported 'not sure' as their COVID-19 circumstance had a significantly lower non-household contact rate than those not self-isolating or shielding. This may have been due to a pause in shielding guidance coinciding with the release of the survey, which may have left participants unsure of their current circumstance.[36–38]

This survey captured the point in time where cases were starting to consistently rise for the first time since March 2020, with the reproduction number estimated to be between 0.8 and 1.1.[15 39–41] The level of social mixing in the UK at the time of this survey enabled epidemic growth.

This study was likely subject to recruitment bias, as the survey was online and open to anyone living in the UK with no active recruitment process. The survey was under-represented by children, teenagers, young adults and the very elderly, as well as ethnic minorities. In particular, under-representation of the very elderly (80+) limited our ability to gain insight into mixing patterns of the age group at highest risk of severe COVID-19 disease. In addition, as we asked participants to report their contact rate, the study may have suffered from recall bias. If a participant reported meeting 15 or more contacts, information was asked about their contacts collectively rather than as individual interactions. When grouping contacts into age groups, participants could select up to '20+' contacts for each age group, which may have led to us underestimating some participant's non-household contact rates (see online supplemental materials). Participants were asked about their current COVID-19 circumstance and contact behaviour for consecutive days (contacts were those made the previous day), which may bias the association of contact rate with COVID-19 circumstance. Comparisons to prepandemic contact levels in the UK are based on social contact studies conducted within the UK prior to 2020, however, these are subject to differences in study population and study design in particular sample distributions and data collection methods.

**Acknowledgements** We would like to thank the participants of the study for providing their time and information, and Prof Julia Gog OBE and Rev Richard Coles for helping to promote the survey.

**Contributors** JREB, CJ and JMR all conceived and designed the study. JREB conducted the analysis and wrote the first draft of the manuscript. All authors edited the manuscript. JMR is guarantor of the article.

**Funding** JREB is supported by a Lancaster University Faculty of Health and Medicine doctoral scholarship. JMR and CJ were supported by UKRI through the JUNIPER modelling consortium (grant number MR/V038613/1).

**Competing interests** None declared.

**Patient and public involvement** Patients and/or the public were not involved in the design, or conduct, or reporting, or dissemination plans of this research.

**Patient consent for publication** Not applicable.

**Ethics approval** This study was approved by the Faculty of Health and Medicine Ethics Committee at Lancaster University (reference FHMREC19135). Participation in the study was voluntary, with each participant (and where appropriate parent or guardian) giving their consent before proceeding. Participants gave informed consent to participate in the study before taking part.

**Provenance and peer review** Not commissioned; externally peer reviewed.

**Data availability statement** Data are available in a public, open access repository.

**ORCID iDs**
Jessica RE Bridgen http://orcid.org/0000-0001-5497-2700
Jonathan M Read http://orcid.org/0000-0002-9697-0962

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
