## [Reviewer comments · BMJ Open]

ARTICLE DETAILS

TITLE (PROVISIONAL)	Social mixing patterns in the UK following the relaxation of COVID-19 pandemic restrictions, July to August 2020: a cross-sectional online survey
AUTHORS	Bridgen, Jessica; Jewell, Christopher; Read, Jonathan

VERSION 1 – REVIEW

REVIEWER	Fang, Chi-Tai National Taiwan University
REVIEW RETURNED	21-Feb-2022

GENERAL COMMENTS	This well-done work provides highly valuable quantitative measurement on social mixing patterns in UK. I have only two minor suggestions: 1. Actually, the UK relaxed pandemic restriction twice, in 2020 and 2021, respectively. Therefore, the title should specify the timing of measurement. For example: "Social mixing pattern in the UK following the relaxation of COVID-19 pandemic restriction, July to August 2020: a cross-sectional online survey".2. A major concern is the claim (in conclusion and the first paragraph of discussion) that "the daily rate of non-household interactions remain lower than pre-pandemic levels", using pre-pandemic POLYMOD study (2005-2006) as the reference. I am not sure whether participants and survey methods are sufficiently similar between the two studies to allow the authors made such claim. Try to transparently specify the finding as "...lower than pre-pandemic levels measured in the POLYMOD study (2005-2006),..." and address the potential difference in participants and survey methods in the limitation section.
---

REVIEWER	Coletti, Pietro Universiteit Hasselt, Censtat
REVIEW RETURNED	24-Feb-2022

GENERAL COMMENTS	The manuscript introduces the study objective, the analysis and the results in a clear and comprehensive way. My only, minor, suggestion is to link its results to similar studies carried out in Europe and worldwide. To this end, a rapid review on social contact patterns during the COVID-19 pandemic (Liu et al, Epidemiology, 2021) and a study performed in Belgium during the same period and a similar epidemiological situation (Coletti et al, Scientific Reports, 2021) could help an interested reader to put the results in perspective. This is indeed a suggestion, as the paper
---

	links the results of the study mostly to other studies in the UK, with only a comparison with Luxembourg being included. I think this manuscript presents the study in a complete way, allowing for reproducing its results. The statistical analysis is sound and fully documented. I therefore think this paper should be accepted.
--	---

VERSION 1 – AUTHOR RESPONSE

Reviewer 1

Prof. Chi-Tai Fang, National Taiwan University

Comments to the Author:

This well-done work provides highly valuable quantitative measurement on social mixing patterns in UK. I have only two minor suggestions:

We thank the reviewer for taking the time respond to our manuscript and for the positive feedback.

Actually, the UK relaxed pandemic restriction twice, in 2020 and 2021, respectively. Therefore, the title should specify the timing of measurement. For example: "Social mixing pattern in the UK following the relaxation of COVID-19 pandemic restriction, July to August 2020: a cross-sectional online survey".

We thank the reviewer for this suggestion. We have changed the title to '*Social mixing pattern in the UK following the relaxation of COVID-19 pandemic restriction, July to August 2020: a cross-sectional online survey*' to better reflect the study period (lines 1-3).

A major concern is the claim (in conclusion and the first paragraph of discussion) that "the daily rate of non-household interactions remain lower than pre-pandemic levels", using pre-pandemic POLYMOD study (2005-2006) as the reference. I am not sure whether participants and survey methods are sufficiently similar between the two studies to allow the authors made such claim. Try to transparently specify the finding as "...lower than pre-pandemic levels measured in the POLYMOD study (2005-2006),..." and address the potential difference in participants and survey methods in the limitation section.

We thank the reviewer for this comment.

We have acknowledged the potential differences in study population and design of pre-pandemic contact studies as a limitation in the discussion. '*Comparisons to pre-pandemic contact levels in the UK are based on social contact studies conducted within the UK prior to 2020, however, these are subject to differences in study population and study design in particular sample distributions and data collection methods.*' (lines 362-365).

We also more clearly refer to 'lower pre-pandemic contact levels' in reference to multiple studies in the abstract and discussion. '*The daily rate of non-household interactions remained lower than pre-pandemic levels measured by other studies, suggesting continued adherence to social distancing guidelines*'. (lines 34-35)

'We found the daily rate of social contact was considerably lower than that measured prior to 2020 in similar but non-identical studies, despite our study period corresponding to a time when the COVID-19 pandemic social distancing restrictions were at their most relaxed during 2020 in the UK.' (lines 286-289)

Reviewer 2

Dr. Pietro Coletti, Universiteit Hasselt

Comments to the Author:

The manuscript introduces the study objective, the analysis and the results in a clear and comprehensive way.

We thank the reviewer for their feedback and positive comments.

My only, minor, suggestion is to link its results to similar studies carried out in Europe and worldwide. To this end, a rapid review on social contact patterns during the COVID-19 pandemic (Liu et al, Epidemiology, 2021) and a study performed in Belgium during the same period and a similar epidemiological situation (Coletti et al, Scientific Reports, 2021) could help an interested reader to put the results in perspective. This is indeed a suggestion, as the paper links the results of the study mostly to other studies in the UK, with only a comparison with Luxembourg being included.

We thank the reviewer for this suggestion. We now include additional references of non-UK contact studies in the discussion. 'Social contact studies outside of the UK also reported low daily contact rates in 2020.' (lines 292-293).

Additional references included: Zhang *et al* (Science, 2020), Coletti *et al* (Scientific Reports, 2021) and Liu *et al* (Epidemiology, 2021).